# The aryl hydrocarbon receptor and interferon gamma generate antiviral states via transcriptional repression

Tonya Kueck[1], Elena Cassella[1], Jessica Holler[2], Baek Kim[2,3], Paul D Bieniasz[1,4]*

[1]Laboratory of Retrovirology, The Rockefeller University, New York, United States; [2]Center for Drug Discovery, The Department of Pediatrics, Emory University, Atlanta, United States; [3]Department of Pharmacy, Kyung Hee University, Seoul, South Korea; [4]Howard Hughes Medical Institute, The Rockefeller University, New York, United States

**Abstract** The aryl hydrocarbon receptor (AhR) is a ligand-dependent transcription factor whose activation induces the expression of numerous genes, with many effects on cells. However, AhR activation is not known to affect the replication of viruses. We show that AhR activation in macrophages causes a block to HIV-1 and HSV-1 replication. We find that AhR activation transcriptionally represses cyclin-dependent kinase (CDK)1/2 and their associated cyclins, thereby reducing SAMHD1 phosphorylation, cellular dNTP levels and both HIV-1 and HSV-1 replication. Remarkably, a different antiviral stimulus, interferon gamma (IFN-γ), that induces a largely non-overlapping set of genes, also transcriptionally represses CDK1, CDK2 and their associated cyclins, resulting in similar dNTP depletion and antiviral effects. Concordantly, the SIV Vpx protein provides complete and partial resistance to the antiviral effects of AhR and IFN-γ, respectively. Thus, distinct antiviral signaling pathways converge on CDK/cyclin repression, causing inhibition of viral DNA synthesis and replication.
DOI: https://doi.org/10.7554/eLife.38867.001

*For correspondence:
pbieniasz@rockefeller.edu

**Competing interests:** The authors declare that no competing interests exist.

## Introduction

Hosts have adopted numerous strategies to hinder the replication of invading bacterial or viral pathogens. Indeed, a diverse set of intrinsic or innate immune defense mechanisms enable the detection and destruction of foreign protein or nucleic acid associated molecular patterns, and the production of antiviral cytokines. The antiviral cytokines induce expression of genes whose products have antiviral activity, facilitate the recruitment of immune cells and activate the adaptive immune system for the establishment of protective immunological memory (*Stetson and Medzhitov, 2006*). Like many viruses, the human and simian immunodeficiency viruses (HIV and SIV) have acquired various mechanisms to defeat intrinsic, innate and adaptive immunity (*Blanco-Melo et al., 2012*; *Malim and Bieniasz, 2012*).

Key components of the innate defenses against viruses are induced by interferons (IFN), that elicit the expression of antiviral interferon-stimulated genes (ISGs) resulting in the so called 'antiviral state' (*Schoggins et al., 2011*; *Kane et al., 2016*). Most attention has been given to how type-I IFNs inhibit viral infection, and a number of proteins with anti HIV-1 activity are known to be type-I ISGs. Conversely, type-II IFN (IFN-γ) is thought to facilitate protection largely through immunomodulatory rather than directly antiviral mechanisms (*Roff et al., 2014*). Nonetheless, IFN-γ production is induced in the initial stages of HIV-1 infection (*Stacey et al., 2009*) and some reports indicate that IFN-γ possesses anti-HIV-1 activity, particularly in macrophages (*Hammer et al., 1986*; *Koyanagi et al., 1988*; *Kornbluth et al., 1989*; *Meylan et al., 1993*). Nevertheless, the molecular

mechanisms underlying inhibition by IFN-γ have been only partly elucidated (*Kane et al., 2016*; *Rihn et al., 2017*).

In addition to cytokines, other endogenous and exogenous substances, modulate the immune responses to viruses. The aryl hydrocarbon receptor (AhR) is a ligand-dependent transcription factor that is activated by a range of organic molecules of endogenous and exogenous origin, including environmental toxins, tryptophan metabolites, and products of the microbiome (*Stockinger et al., 2014*; *Zhang et al., 2017*). In response to ligand activation, AhR translocates from the cytoplasm into the nucleus where it induces the transcription of numerous target genes including detoxifying monooxygenases CYP1A1 and CYP1B1, as well as its own negative regulator, the AhR repressor (AHRR) (*Stockinger et al., 2014*). AhR activation can have a variety of effects on cell physiology, impacting proliferation and differentiation (*Quintana et al., 2008*; *Veldhoen et al., 2008*). In the immune system, AhR activation has modulatory activities, including exerting effects on cytokine secretion (*Stockinger et al., 2014*; *Gutiérrez-Vázquez and Quintana, 2018*). Additionally, AhR was recently identified as a pattern-recognition receptor for bacterial pigments (*Moura-Alves et al., 2014*). AhR signaling can reduce type-I IFN antiviral immune responses (*Yamada et al., 2016*) and AhR expression is induced in astrocytes by type-I IFN, leading to the suppression of inflammation in the central nervous system (*Rothhammer et al., 2016*).

Indoleamine 2,3-dioxygenase 1 (IDO1), is among the most strongly upregulated genes following IFN-γ stimulation. Interestingly, IDO1 catalyzes the initial and rate limiting step in conversion of L-tryptophan into kynurenine (*Hayaishi, 1976*), is upregulated during HIV-1 infection or IFN-γ stimulation (*Favre et al., 2010*) and can block the replication of HIV-1 and other viruses through the inhibition of viral protein production by L-tryptophan depletion (*Kane et al., 2016*; *Mao et al., 2011*; *Obojes et al., 2005*). IDO1 is also thought to be responsible for the generation of tryptophan-derived endogenous AhR ligands (*Zelante et al., 2014*; *Romani et al., 2014*), raising the possibility of cross-talk between IFN-γ and AhR induced pathways. Nevertheless, while the effects of AhR activation on regulation of immune responses has been frequently studied (*Stockinger et al., 2014*; *Gutiérrez-Vázquez and Quintana, 2018*), it is not known whether or how AhR activation, and the genes that it induces, affect the replication of viruses.

Here, we investigated whether and how AhR and IFN-γ affect HIV-1 replication in primary cells. We show that AhR and IFN-γ stimulation have little or no effect on HIV-1 replication in T-cells, but both profoundly inhibit HIV-1 replication in macrophages. While AhR agonists and IFN-γ activate the transcription of sets of genes that are nearly completely distinct from each other, we find that both stimuli downregulate the transcription of CDK1, CDK2 and associated cyclins. These transcriptional suppression activities of AhR and IFN-γ reduce the phosphorylation of SAMHD1, activating its dNTP triphosphohydrolase (dNTPase) activity and reduce cellular dNTP levels. Concordantly, the replication of a completely distinct virus that depends on dNTPs for replication, namely herpes simplex virus-1 (HSV-1), is also inhibited by both AhR activation and IFN-γ. SIV Vpx, confers resistance to inhibition by AhR agonists and IFN-γ. Thus, two distinct signaling pathways induce an antiviral state through the activation of SAMHD1 and consequent lowering the levels of cellular dNTPs required for viral DNA synthesis.

## Results

### AhR activation inhibits HIV-1 replication in human macrophages

While immune modulation by AhR ligands has been extensively studied (*Stockinger et al., 2014*; *Gutiérrez-Vázquez and Quintana, 2018*), the impact of AhR activation on intrinsic host defenses against viral infections is unknown. To determine the impact of AhR activation on HIV-1 replication, we infected primary cells with HIV-1$_{NLYU2}$, a cloned HIV-1 bearing the R5-tropic envelope protein from the YU2 isolate or HIV-1$_{89.6}$, a cloned primary dual-tropic HIV-1 isolate, 89.6. Replication of HIV-1$_{NLYU2}$ or HIV-1$_{89.6}$ in CD4+ T cells or unfractionated peripheral blood mononuclear cells (PBMC) was unaffected by treatment with the naturally occurring AhR agonist 6-Formylindolo(3,2-b) carbazole (FICZ) or the synthetic AhR antagonist CH-223191 (*Figure 1A* and *Figure 1—figure supplement 1A–D and I*). Conversely, replication of both HIV-1 strains in macrophages was inhibited by 10- to 1000-fold, depending on the donor and virus strain used (*Figure 1B* and *Figure 1—figure supplement 1E–H and I*). In macrophages from some donors, the AhR antagonist enhanced HIV-1

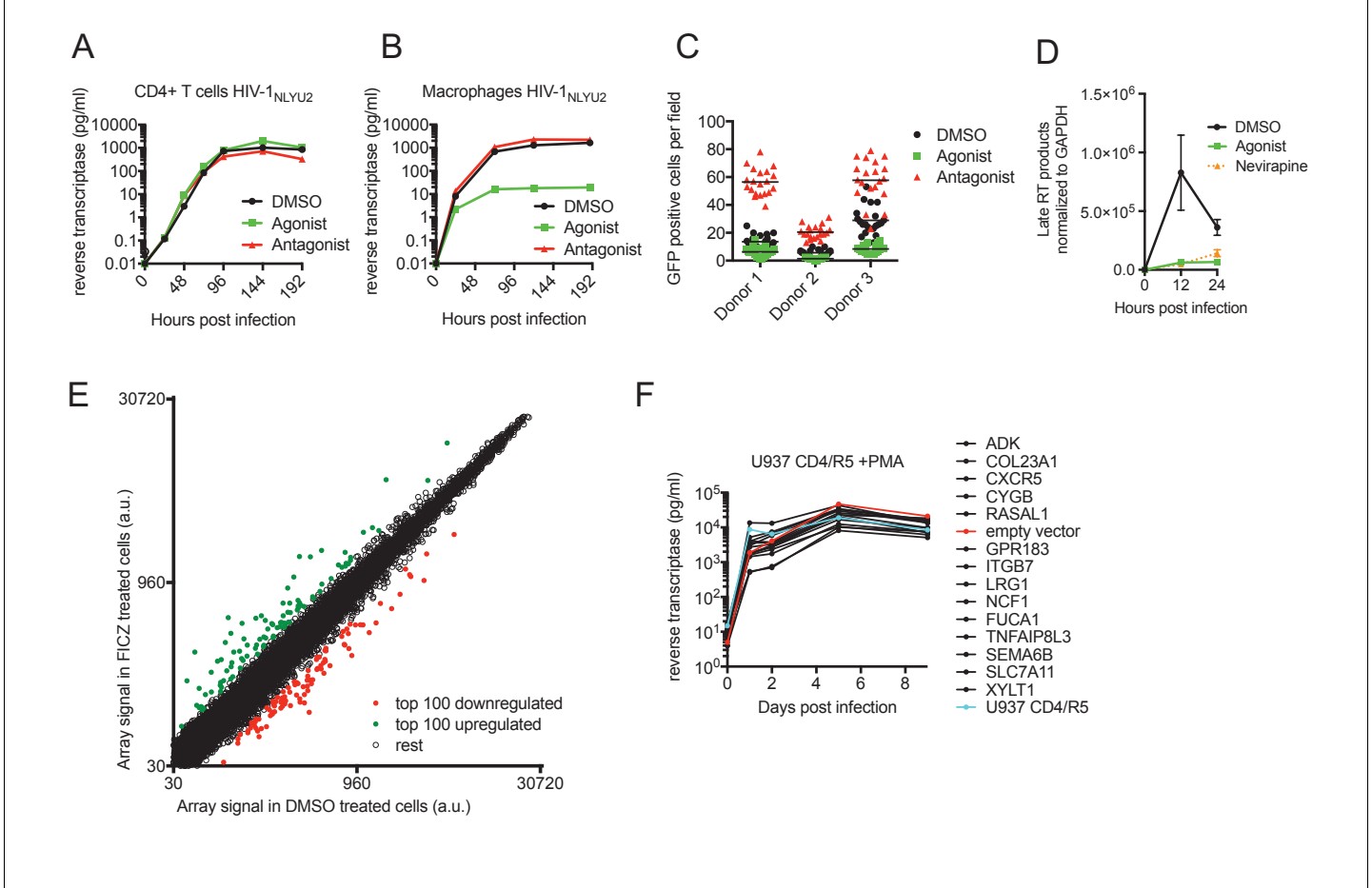

**Figure 1.** AhR activation inhibits HIV-1 replication in macrophages. (A and B) HIV-1$_{NLYU2}$ replication in CD4+ T cells (A) or macrophages (B) treated with AhR agonist or antagonist. Representative of at least three different donors. (C) Numbers of infected macrophages after HIV-1$_{NLYU2}$-GFP single-round infection. Each symbol represents a single field in which numbers of infected cells were counted. (D) Quantitative PCR analysis of late HIV-1 DNA (late reverse transcript (RT)) abundance in macrophages. Error bars represent standard deviation, n = 4. (E) Microarray analysis of RNA extracted from macrophages treated with AhR agonist or control. The array signal is plotted in arbitrary units (a.u.). (F) HIV-1$_{NLYU2}$ replication in a PMA-differentiated monocytic cell line (U937 CD4/R5) transduced with a lentiviral vectors (CSIB) expressing the indicated genes that were upregulated in response to the AhR agonist (from panel (E)). Representative example of 2 independent experiments.

DOI: https://doi.org/10.7554/eLife.38867.002

The following figure supplements are available for figure 1:

**Figure supplement 1.** Effect of AhR activation on HIV-1 replication.

DOI: https://doi.org/10.7554/eLife.38867.003

**Figure supplement 2.** Effect of AhR activation on STAT1, cell-cycle and gene expression in macrophages.

DOI: https://doi.org/10.7554/eLife.38867.004

replication (*Figure 1—figure supplement 1E*). Donor variation in the ability of macrophages to support HIV-1 replication is a well known but unexplained phenomenon, we noted that the levels of CD163 expression levels varied from donor to donor, while other macrophage markers CD14, CD206 and CD80 were consistently expressed (*Figure 1—figure supplement 2A*).

To determine at which step the AhR agonist induced a block in the HIV-1 replication cycle, we executed single cycle infection experiments using a VSV-G pseudotyped HIV-1-GFP reporter virus, that expresses GFP in infected cells. There were ~3 fold fewer GFP+ infected cells when macrophages were pretreated with FICZ (AhR agonist) and ~3 fold more GFP+ infected cells when macrophages were pretreated with CH-223191 (AhR antagonist), compared to the control-treated macrophages (*Figure 1C*). Thus, AhR activation blocked virus replication at early step in the replication cycle, prior to the onset of early gene expression. Next, we used quantitative PCR assays to

monitor HIV-1 DNA synthesis during HIV-1 infection of macrophages, and found that viral DNA synthesis was profoundly inhibited by AhR activation. Indeed, levels of late reverse transcripts in infected FICZ-treated macrophages were similar to those in macrophages treated with the reverse transcription inhibitor nevirapine (*Figure 1D*). Taken together, these data indicated that AhR activation in human macrophages imparts a block to HIV-1 replication that is imposed before or during reverse transcription.

AhR activation can have a variety of effects on cells that could plausibly lead to inhibition of HIV-1 infection, including STAT1 activation (that might active expression of antiviral genes) or inhibition of cell cycle progression (*Kimura et al., 2008*). However, we found that AhR activation in macrophages did not induce STAT1 phosphorylation, whereas control IFN-γ-treated macrophages exhibited increased STAT1 phosphorylation (*Figure 1—figure supplement 2B*). Moreover, the differentiated macrophages used in these experiments were found to be exclusively in G0/G1 phases of the cell cycle a state that was unaffected by AhR agonist or antagonist treatment (*Figure 1—figure supplement 2C*). In an attempt to identify candidate effectors of the apparent antiviral action of AhR, we used microarrays to measure changes in mRNA levels induced by treatment of human macrophages with the AhR agonist (*Figure 1E*, *Figure 1—figure supplement 2D,E* and *Supplementary file 1*). The top fourteen overlapping genes that were represented in AhR agonist-upregulated genes in macrophages from at least two donors were selected. These fourteen genes were introduced into the monocyte cell line U937 using a retroviral vector, and the cells treated with PMA to induce differentiation into a macrophage-like state prior to infection with HIV-1. However, spreading HIV-1 replication experiments in these cells did not reveal antiviral activity for any of the genes tested (*Figure 1F*).

## AhR activation represses the expression of many genes, including CDK1,2 and associated cyclins

AhR is well known to activate the transcription of target genes. However, in addition to the numerous genes that were upregulated in macrophages by the AhR agonist, we noticed that the expression of a number of genes appeared to be repressed (*Figure 1E*, *Figure 1—figure supplement 2D,E* and *Supplementary file 2*). An analysis of the cellular and molecular functions associated with the 100 most downregulated genes revealed that some candidate genes were likely important for the cell-cycle progression, putatively acting on cell-cycle checkpoints or the G1/S transition (*Supplementary file 3*). Among these downregulated genes, cyclin-dependent kinase CDK1 (also known as CDC2), CDK2 and associated cyclins A2 and E2, were present in the lists of 100 most repressed genes from three different macrophage donors (*Figure 1E*, *Figure 1—figure supplement 2D,E* and *Supplementary file 2*), and were studied in more detail. RT-PCR assays confirmed the that mRNA levels for these genes were reduced by 2.5-fold (CDK2) to 20-fold (CDK1) in AhR-activated macrophages (*Figure 2A*), while a control mRNA encoding CYP1B1 that is known to be induced by AhR activation (reviewed by *Stockinger et al., 2014*), was upregulated by ~4 fold. Western blot analysis also showed that reduction of CDK1, CDK2 and cyclin A2 mRNA levels was also reflected as reduced protein levels, while cyclin E2 protein levels were maintained (*Figure 2B*). Conversely, in human PBMCs and CD4+ T cells, where the antiviral action of the AhR agonist was not observed (*Figure 1—figure supplement 1A–D*), CDK1/2 protein levels remained unchanged upon AhR activation, whether or not the cells were activated with PHA (*Figure 2—figure supplement 1A,B*).

## Repression of CDK1/2 and associated cyclins reduces SAMHD1 phosphorylation and cellular dNTP levels

One of the substrates of CDK1 and CDK2 that was potentially responsible for macrophage specific antiviral effect of their downregulation was the antiviral enzyme, SAM domain and HD domain-containing protein 1 (SAMHD1). Indeed, SAMHD1 has been shown to inhibit HIV-1 and SIV reverse transcription by reducing the levels of dNTPs in dendritic cells, monocytes and macrophages via its dNTP triphosphohydrolase (dNTPase) activity (*Hrecka et al., 2011*; *Laguette et al., 2011*; *Yan et al., 2013*; *DeLucia et al., 2013*). Crucially, SAMHD1 activity is regulated by CDK1/2, with phosphorylation causing inhibition of SAMHD1 tetramerization which in turn reduces dNTPase activity (*Hansen et al., 2014*; *Ji et al., 2013*).

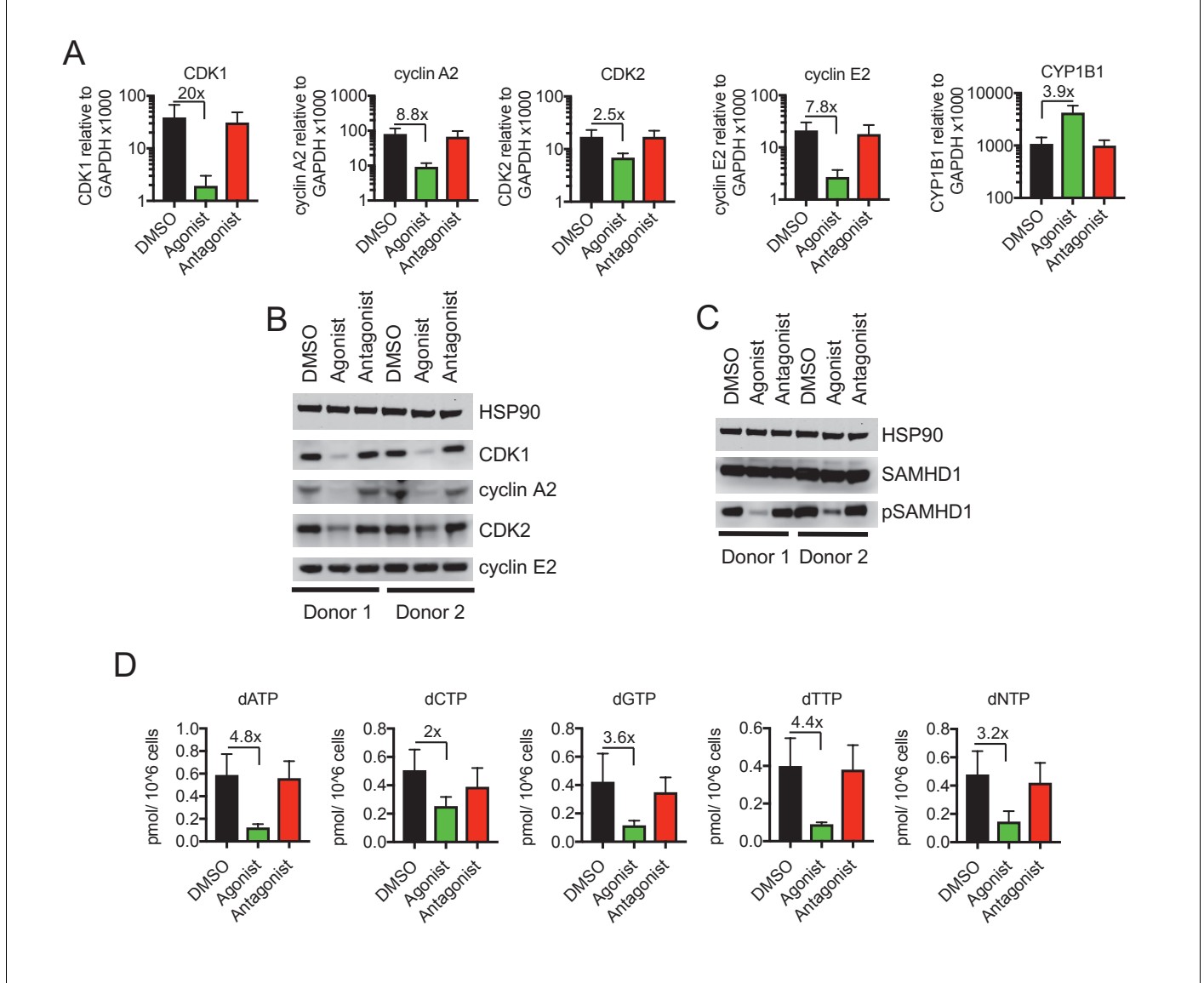

**Figure 2.** Reduced CDK/cyclin, phospho-SAMHD1 and dTNP levels in macrophages following activation of AhR. (A) Quantitative PCR analysis of CDK1, cyclin A2, CDK2, cyclin E2 and CYP1B1 mRNA levels in macrophages treated with carrier (DMSO), AhR agonist or AhR antagonist. Values indicate fold-change, error bars represent standard deviation of data from five different donors. (B) Western blot analysis of HSP90 (loading control) CDK1, cyclin A2, CDK2, cyclin E2 protein levels in macrophages treated with carrier (DMSO), AhR agonist or antagonist. (C) Western blot analysis of HSP90 (loading control), total SAMHD1 levels, and phosphorylated SAMHD1 levels in macrophages treated with carrier (DMSO), AhR agonist or antagonist. (D) Analysis of dNTP levels in macrophages treated with AhR agonist or antagonist. Error bars represent standard deviation of data from three different donors, numbers indicate fold change.

DOI: https://doi.org/10.7554/eLife.38867.005

The following figure supplement is available for figure 2:

**Figure supplement 1.** Effects of AhR activation in PBMC and purified CD4+ T cells.

DOI: https://doi.org/10.7554/eLife.38867.006

Western blot analyses revealed that SAMHD1 protein levels in macrophages were unaltered by AhR activation. Notably however, the degree to which SAMHD1 was phosphorylated was clearly decreased when macrophages were treated with the AhR agonist (*Figure 2C*), consistent with the finding that CDK1/2 mRNA and protein levels were decreased (*Figure 2A,B*). Conversely, AhR activation had no effect on SAMHD1 protein levels or the degree to which SAMHD1 was phosphorylated in PBMCs and CD4+ T cells (*Figure 2—figure supplement 1A,B*).

CDK1/2 mediated SAMHD1 phosphorylation at position T592 impairs tetramerization and dNTPase activity, causing elevated intracellular dNTP concentrations in cycling cells (*Cribier et al., 2013*; *White et al., 2013*). We employed a previously described primer extension assay (*Diamond et al., 2004*) to measure dNTP levels in AhR agonist treated macrophages. Consistent with the effects of AhR activation on CDK1/2/cyclin levels, and the phosphorylation status of SAMHD1, we found that levels of all four dNTPs in macrophages were decreased, by a mean of 3.2-fold, following AhR activation (*Figure 2D*).

## AhR activation inhibits HSV-1 replication in macrophages

Large double-stranded DNA (dsDNA) viruses, including herpes simplex viruses (HSV), also infect non-dividing cells such as macrophages and, like HIV-1, require cellular dNTPs for viral DNA synthesis. Indeed, SAMHD1 has been shown to inhibit the replication of dsDNA viruses through cellular dNTP depletion (*Hollenbaugh et al., 2013*; *Kim et al., 2013*). We infected AhR activated or control human monocyte-derived macrophages with an HSV-1 strain (KOS) and monitored progeny viral yield and the expression of HSV-1 glycoprotein D (gD), a structural component of the HSV envelope. Lower levels of gD protein were observed in AhR activated macrophages compared to control cells,

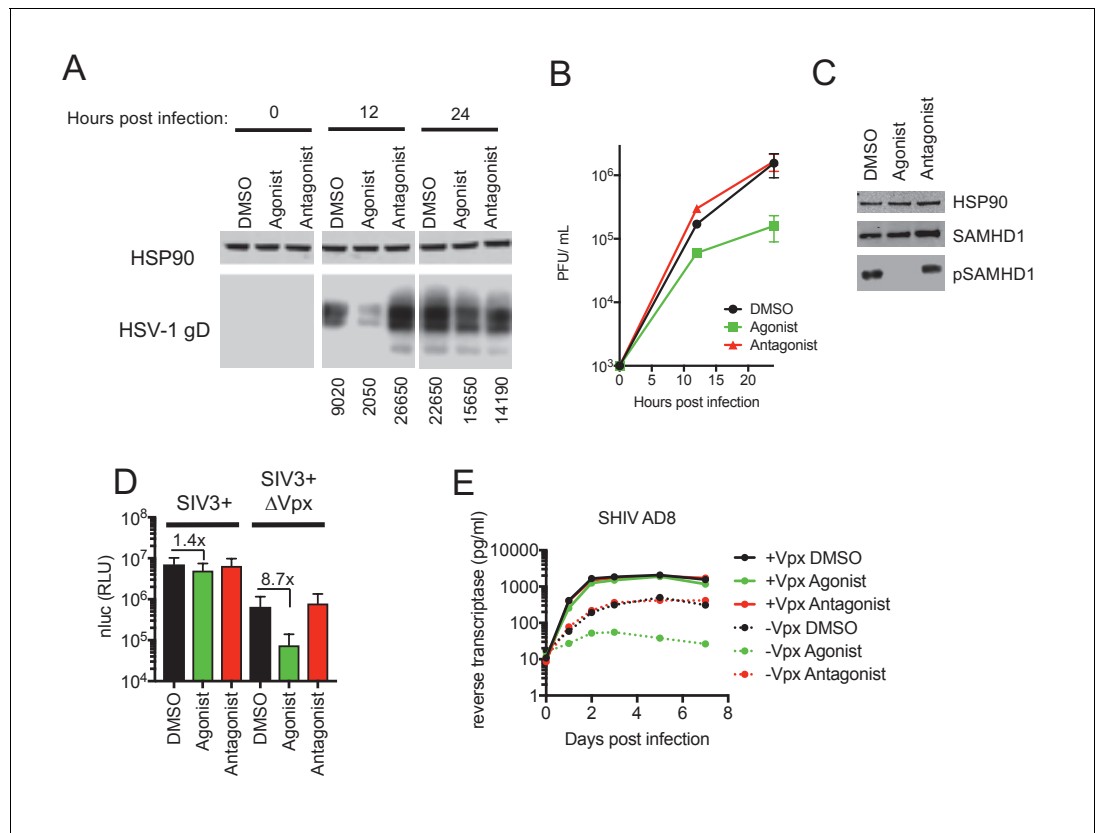

**Figure 3.** AhR activation inhibits HIV-1, and HSV-1 replication, but SIVmac Vpx confers resistance. (**A**) Western blot analysis of HSV-1 glycoprotein D (gD) levels over time in infected macrophages treated with AhR agonist or antagonist. Numbers represent relative band intensities. Representative of 2 different donors. (**B**) HSV-1 infectious virion yield during replication in macrophages treated as in (**A**). Error bars represent standard deviation, n = 2. (**C**) Western blot analysis of SAMHD1 and phospho-SAMHD1 levels in HSV-1 infected macrophages treated with AhR agonist or antagonist. Representative of 2 different donors. (**D**) Single round virus replication in macrophages treated with SIV3+ or SIV3+ ΔVpx virus like particles and AhR agonist or antagonist before infection with HIV-1$_{NLYU2}$-nluc. Error bars represent standard deviation of data from six different donors, numbers indicate fold change. (**E**) Supernatant reverse transcriptase levels in macrophages treated with AhR agonist or antagonist and infected with chimeric SHIV AD8 expressing or not expressing the Vpx protein. Representative of 3 different donors.

DOI: https://doi.org/10.7554/eLife.38867.007

The following figure supplement is available for figure 3:

**Figure supplement 1.** Vpx-induced degradation of SAMHD1 abolishes AhR-mediated inhibition in macrophages.
DOI: https://doi.org/10.7554/eLife.38867.008

with the most pronounced differences occurring at 12 hr post infection (*Figure 3A*). Measurement of plaque forming units (PFU) in culture supernatants revealed that progeny viral yield was decreased in AhR activated macrophages by 12-fold at 24 hr post infection (*Figure 3B*). Notably, HSV-1 infection did not appear to affect the reduction in phosphorylation of SAMHD1 that results from AhR activation (*Figure 3C*). Thus, these data suggest that AhR activation may confer broad inhibition of viruses whose replication depends on DNA synthesis.

## SIV$_{mac}$Vpx abolishes AhR induced inhibition of HIV-1 and SHIV replication

The viral accessory protein Vpx protein is present in HIV-2 and many SIVs, and is incorporated into virions. Vpx facilitates HIV-2 and SIV infection of macrophages by inducing the proteasomal degradation of SAMHD1 (*Hrecka et al., 2011*; *Laguette et al., 2011*). If the block in HIV-1 infection induced by AhR activation in macrophages was primarily caused by SAMHD1 dephosphorylation and consequent dNTP depletion, we reasoned that infection in the presence of Vpx should reduce or abolish the inhibitory effect of AhR activation. To test this idea, we used Vpx-containing and control SIV$_{mac251}$ virus like particles (SIV3+ and SIV3+ ΔVpx) (*Goujon et al., 2013*) respectively, and treated macrophage target cells to induce SAMHD1 degradation (*Figure 3—figure supplement 1A*).

In macrophages treated with particles lacking Vpx (SIV3+ ΔVpx), infection using a HIV-1$_{NLYU2}$-nluc reporter virus was inhibited by ~9 fold by the AhR agonist (*Figure 3D*). Conversely, infection of cells depleted of SAMHD1 by treatment with Vpx containing particles (SIV3+) was both enhanced compared to untreated macrophages and was nearly completely resistant to inhibition by AhR activation (*Figure 3D*). Next we made use of a hybrid simian-human immunodeficiency virus (SHIV) encoding a macrophage-tropic HIV-1 envelope protein in an otherwise SIV$_{mac239}$ background (SHIV AD8) (*Del Prete et al., 2017*) along with a Vpx-deleted counterpart, to measure effects of AhR activation and Vpx spreading replication. As was the case for HIV-1, the replication of SHIV AD8 lacking Vpx was inhibited by 10 to 100-fold by AhR activation, depending on the macrophage donor (*Figure 3E* and *Figure 3—figure supplement 1B–D*). Strikingly however, SHIV AD8 encoding an intact Vpx protein was completely resistant to the effects of AhR activation in multiple donors. Together, these results indicate that the presence of Vpx counteracts the effects of AhR activation and suggest that most, if not all, of the antiviral effect of AhR activation is mediated through SAMHD1.

## IFN-γ inhibits HIV-1 replication in human macrophages independently of IDO1 and AhR

IFN-γ has been shown to induce both early and late blocks to HIV-1 infection in CD4+ T cells, macrophages and a number of commonly used cell lines (*Kane et al., 2016*; *Hammer et al., 1986*; *Koyanagi et al., 1988*; *Kornbluth et al., 1989*; *Meylan et al., 1993*; *Rihn et al., 2017*). The mechanisms underlying this inhibition are not well understood. Nevertheless, IFN-γ has been shown to cause a strain-specific, envelope dependent late block in some cell lines (*Rihn et al., 2017*) and, an IDO1-induced, tryptophan depletion-dependent block in others (*Kane et al., 2016*). However, the mechanisms underlying the inhibition of HIV-1 replication in primary HIV-1 target cells have not been uncovered. Given that IDO1 is known to be upregulated by IFN-γ and is thought to be a key source of AhR agonists (*Zelante et al., 2014*; *Romani et al., 2014*), we investigated how IFN-γ inhibited HIV-1 replication, in parallel to our studies of AhR agonist mediated inhibition.

First, we determined the potency with which IFN-γ inhibited HIV-1 replication in macrophages and PBMC. Levels of HIV-1$_{NLYU2}$ spread in macrophages at 7 days post infection were reduced by IFN-γ in a dose dependent manner, up to >400 fold by treatment with 1000 U/ml of IFN-γ (*Figure 4A* and *Figure 4—figure supplement 1A*). Conversely, spread in PBMCs was inhibited by a comparatively modest 7-fold, dependent on the particular donor (*Figure 4B* and *Figure 4—figure supplement 1B*). Clearly IFN-γ was a much more potent inhibitor of HIV-1 replication in macrophages than in PBMCs (*Figure 4C*).

To identify candidate effectors responsible for the IFN-γ mediated block in macrophages, we compared mRNA transcriptomes in cells that were untreated or treated with IFN-γ (*Figure 4D*, *Figure 4—figure supplement 1C,D* and *Supplementary file 4*). The most highly induced gene in all three donors tested was IDO1, which was upregulated between 160- to 250-fold by IFN-γ

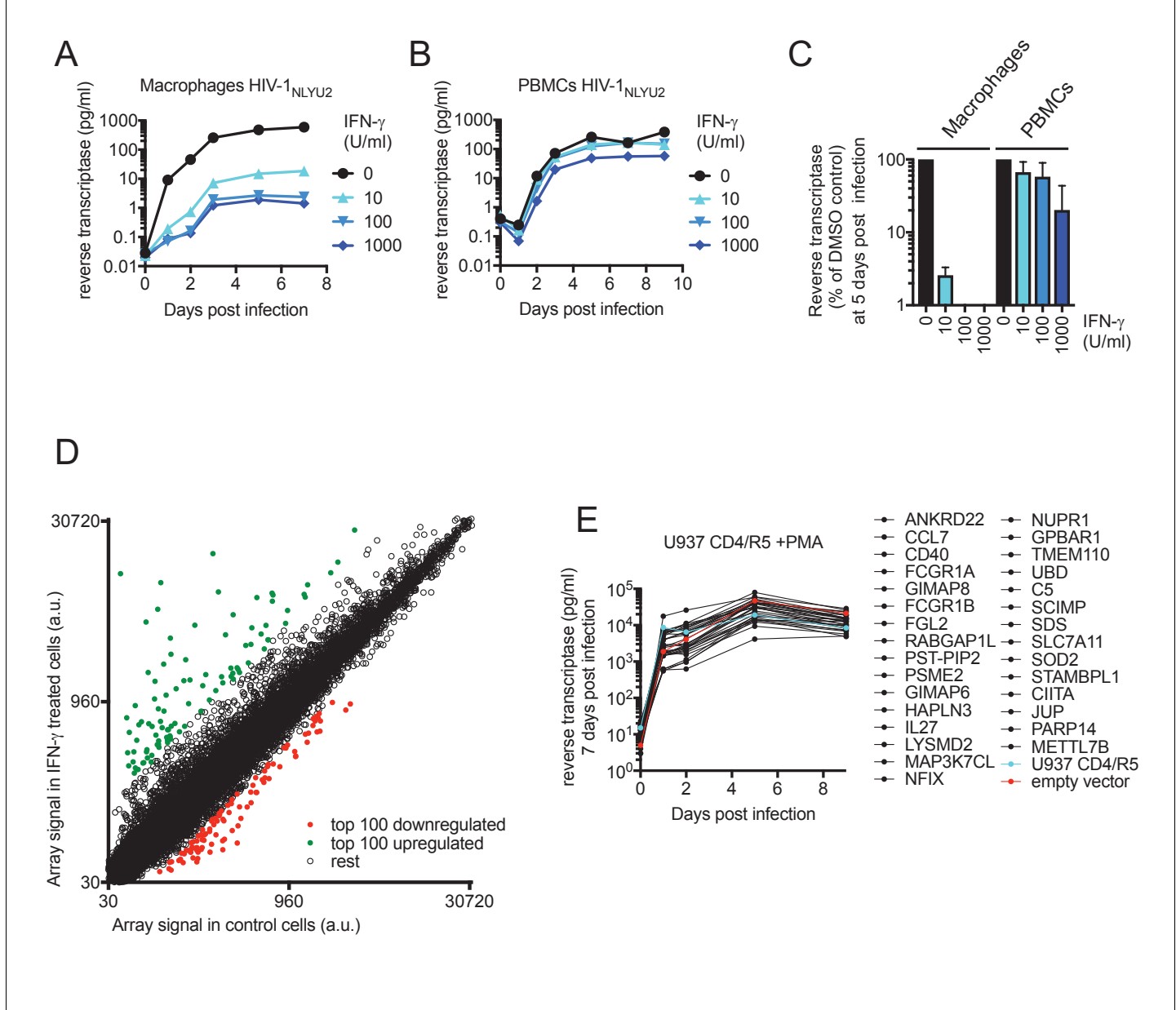

**Figure 4.** IFN-γ inhibits HIV-1 replication in macrophages. (**A and B**) HIV-1_NLYU2 replication in macrophages (**A**) or PBMCs (**B**) treated with increasing concentrations of IFN-γ. Representative of 3 different donors. (**C**) Comparison of HIV-1 supernatant reverse transcriptase levels, at five days after infection of macrophages or PBMCs treated with increasing concentrations of IFN-γ from (**A and B**). Error bars represent standard deviation of data from at least three different donors. (**D**) Microarray analysis of RNA extracted from macrophages treated with IFN-γ or control. The array signal is plotted in arbitrary units (a.u.). (**E**) HIV-1_NLYU2 replication in a PMA treated monocytic cell line (U937 CD4/R5) transduced with lentiviral vectors (CSIB) expressing the indicated upregulated genes from (**D**). Representative example of 2 independent experiments.

DOI: https://doi.org/10.7554/eLife.38867.009

The following figure supplements are available for figure 4:

**Figure supplement 1.** Effect of IFN-γ on HIV-1 replication and microarray analysis of IFN-γ-induced changes in mRNA levels.

DOI: https://doi.org/10.7554/eLife.38867.010

**Figure supplement 2.** IDO1-mediated Tryptophan catabolism does not mediate the IFN-γ induced replication block.

DOI: https://doi.org/10.7554/eLife.38867.011

(*Supplementary file 4* and *Figure 4—figure supplement 2A*). Crucially however, media supplementation with tryptophan, or the IDO1 inhibitor 1-Methyl-L-tryptophan had no effect on HIV-1 replication (*Figure 4—figure supplement 2B,C*) or AhR activation (*Figure 4—figure supplement 2D,E*), suggesting that the IFN-γ mediated block to HIV-1 in macrophages is different to that resulting from L-tryptophan depletion in T cell lines (*Kane et al., 2016*). Moreover, this finding also suggested that IFN-γ/IDO1 driven generation of kynurenine or other AhR ligands from tryptophan metabolites was not responsible for the inhibition of HIV-1 via AhR activation. Indeed, RT-PCR and microarray analyses revealed that the AhR target gene CYP1B1 was not induced by IFN-γ (*Figure 5A*). In fact, the sets of genes induced by AhR activation and IFN-γ treatment were largely non-overlapping (*Supplementary file 1* and *Supplementary file 4*). To determine which IFN-γ-induced genes might be responsible for inhibition, we selected 30 candidate genes that were upregulated in at least three donors (*Supplementary file 4*) and expressed them in the monocyte cell line U937. Thereafter, cells were differentiated into a macrophage-like state using PMA, and tested for their ability to support HIV-1 replication. Notably, none of the genes tested had a major effect on HIV-1 replication (*Figure 4E*).

## IFN-γ downregulates CDK1, CDK2 and cyclins and thereby activates SAMHD1 in macrophages

Despite the fact that IFN-γ induced a different transcriptional signature to AhR activation, we noticed that IFN-γ treatment, like AhR activation, caused reduction in the levels of many mRNAs in macrophages (*Figure 4D*, *Figure 4—figure supplement 1C,D* and *Supplementary file 5*). Given the above findings with AhR, we considered whether the downregulation of particular genes by IFN-γ might play a role in HIV-1 inhibition. Remarkably, and similar to the results obtained with AhR, microarray analyses suggested that CDK1 and cyclin A2 and E2 were among the most strongly downregulated genes following IFN-γ treatment of macrophages (*Supplementary file 5*). RT-PCR analyses confirmed that CDK1, CDK2 and associated cyclin A2 and E2 mRNA levels were reduced to varying degrees (2.5-fold to 68-fold) upon IFN-γ treatment (*Figure 5A*). Moreover, western blot analyses indicated that CDK1, CDK2 and cyclin A2 protein levels were reduced in IFN-γ activated macrophages (*Figure 5B*), to a similar degree as in AhR activated cells (*Figure 2B*).

These data suggested that even though IFN-γ treatment does not cause AhR activation, the antiviral pathways triggered by IFN-γ and AhR activation nevertheless converge, in that both treatments lead to transcriptional silencing of CDK1, CDK2 and associated cyclins. Therefore, we investigated whether SAMHD1 might be required for the generation of an IFN-γ induced antiviral state in macrophages. We assessed whether the IFN-γ-mediated reduced levels of CDK1/2 and cyclins observed in IFN-γ treated cells affected the phosphorylation and activity of SAMHD1. Indeed, IFN-γ treatment reduced levels of phosphorylated SAMHD1 while leaving the total levels of SAMHD1 unaffected (*Figure 5C*). There was a commensurate decrease in the level of dNTPs (*Figure 5D*) that was similar in magnitude (2.1-fold to 5.3-fold) to the decrease observed upon AhR activation (*Figure 2D*).

## IFN-γ inhibits HIV-1 and HSV-1 replication in macrophages, but SIV Vpx confers resistance to IFN-γ

Consistent with the notion that dNTP depletion is a major contributor to the antiviral activity of IFN-γ in macrophages, we found that HSV-1 replication was inhibited by IFN-γ therein, as was the case with AhR activation. Indeed, IFN-γ treatment resulted in reduced levels of the HSV-1 envelope protein gD in infected cells, as well as reduced yield (30-fold) of infectious progeny virions (*Figure 6A, B*), as was the case with AhR activation (*Figure 3A,B*).

If SAMHD1 dephosphorylation and consequent dNTP depletion contributed to the observed block in HIV-1 infection induced by IFN-γ, then Vpx should confer at least some level of resistance. In single cycle infection experiments, in which macrophages were infected with an HIV-1$_{NLYU2}$-nluc reporter virus in the presence of Vpx-null SIV particles, treatment with IFN-γ resulted in a 50-fold inhibition of infection. Conversely, in the presence of Vpx-carrying SIV VLPs, infection by HIV-1$_{NLYU2}$-nluc was increased compared to Vpx-null SIV particles, and was nearly completely resistant to inhibition by IFN-γ (*Figure 6C*). This finding strongly suggested that SAMHD1 is required for the majority of the effect of IFN-γ in single cycle infection assays. In spreading replication experiments with SHIV AD8 or its Vpx-deleted counterpart, conducted in the presence of increasing concentrations of IFN-

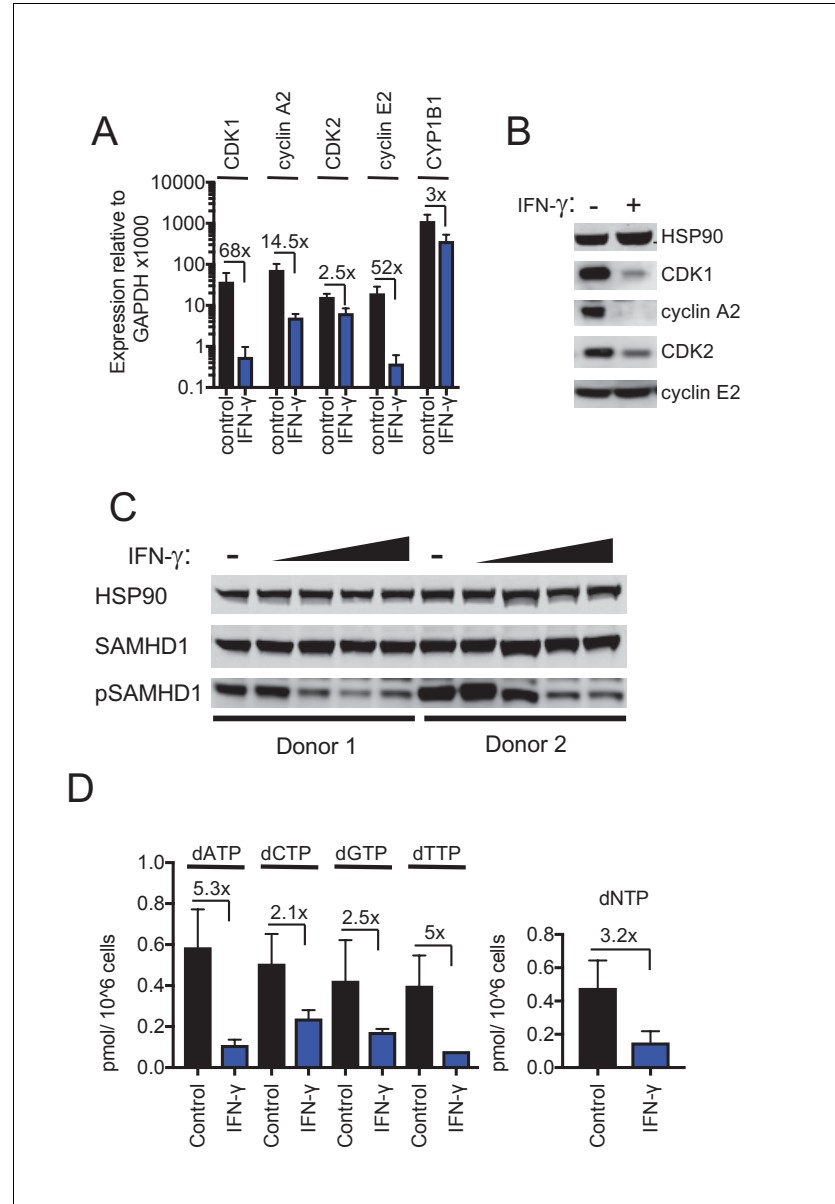

**Figure 5.** IFN-γ-mediated inhibition of viral replications is mechanistically similar to AhR. (**A**) Quantitative PCR analysis of CDK1, cyclin A2, CDK2, cyclin E2 and CYP1B1 mRNA levels in macrophages that were untreated or treated with IFN-γ. Numbers indicate fold-change, error bars represent standard deviation of data from five different donors. (**B**) Western blot analysis of HSP90 (loading control) CDK1, cyclin A2, CDK2, cyclin E2 protein levels in macrophages treated with carrier (DMSO), AhR agonist or antagonist. (**C**) Western blot analysis of HSP90 (loading control), total SAMHD1 levels, and phosphorylated SAMHD1 levels in macrophages treated with increasing concentrations of IFN-γ. Representative of 3 different donors. (**D**) Analysis of dNTP levels in macrophages treated with IFN-γ. Error bars represent standard deviation of data from three different donors. Numbers indicate fold change.
DOI: https://doi.org/10.7554/eLife.38867.012

γ, Vpx again caused substantial, albeit incomplete, resistance to the antiviral effects of IFN-γ (*Figure 6D*). These data suggest that the IFN-γ induced antiviral state in macrophages is largely due to reduced SAMHD1 phosphorylation. However, IFN-γ treatment could induce additional blocks to virus replication, resulting in residual sensitivity to IFN-γ in a Vpx-expressing SHIV (*Figure 6C,D*).

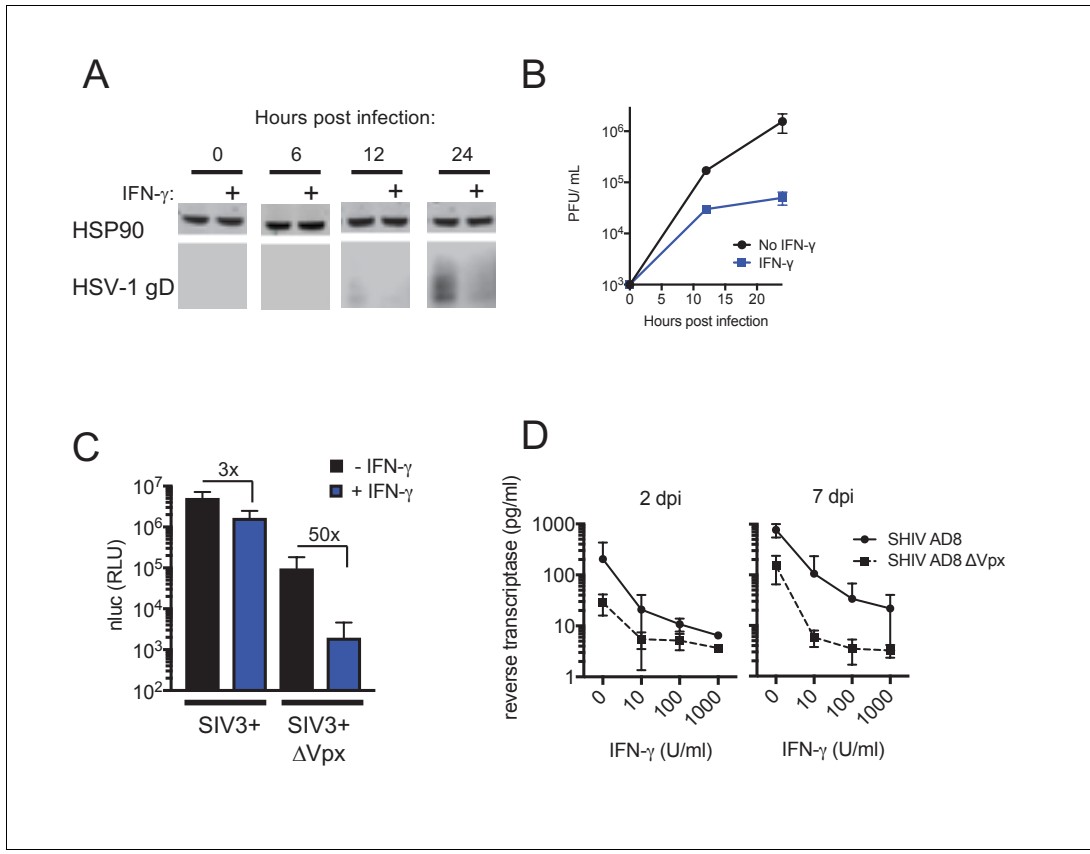

**Figure 6.** IFN-γ inhibits HIV-1 and HSV-1 replication, but SIV$_{mac}$ Vpx confers resistance. (A) Western blot analysis of HSV-1 glycoprotein D (gD) over time in infected macrophages that were either untreated or treated with IFN-γ. (B) HSV-1 infectious virion yield during replication in macrophages treated as in (A). Error bars represent standard deviation of two independent experiments. (C) Single round infection in macrophages treated with SIV3+ or SIV3+ ΔVpx virus like particles and IFN-γ before infection with HIV-1$_{NLYU2}$-nluc. Numbers indicate fold change. Error bars represent standard deviation of data from three different donors. (D) SHIV AD8 and SHIV AD8 ΔVpx reverse transcriptase levels in infected macrophages treated with increasing concentrations of IFN-γ. Error bars represent standard deviation of data from two different donors.
DOI: https://doi.org/10.7554/eLife.38867.013

## CDK1 mRNA turnover is unaffected by AhR and IFN-γ

The antiviral states induced by AhR agonists and IFN-γ were ultimately manifested, in both cases, as a reduction of the steady state levels of CDK1, CDK2, and associated cyclin mRNAs, revealing convergence in the mechanism by which they cause SAMHD1 activation and dNTP depletion. Any reduction in mRNA steady state level could, in principle, be due to reduced mRNA synthesis (transcription) or increased turnover (degradation). To distinguish between these possibilities, we focused on CDK1, which is both primarily responsible for SAMHD1 phosphorylation and showed the greatest reduction in mRNA levels in response to AhR activation or IFN-γ signaling. In untreated cells, CDK1 mRNA was depleted with an apparent half-life of 2.05 hr following actinomycin D treatment (*Figure 7A,C*). Conversely, and as expected, control mRNAs encoding TNF-α and IL-1β that are known to be rapidly turned over (*Mahmoud et al., 2014*), were quickly depleted following actinomycin D treatment, with a measured half-life of less than one hour in both cases (*Figure 7A*). Notably, there was little or no change in the turnover of CDK1 mRNA following AhR activation or IFN-γ treatment (*Figure 7B,C*), strongly suggesting that the downregulation of CDK1 mRNA following these stimuli is caused by reduced synthesis rather than increased degradation.

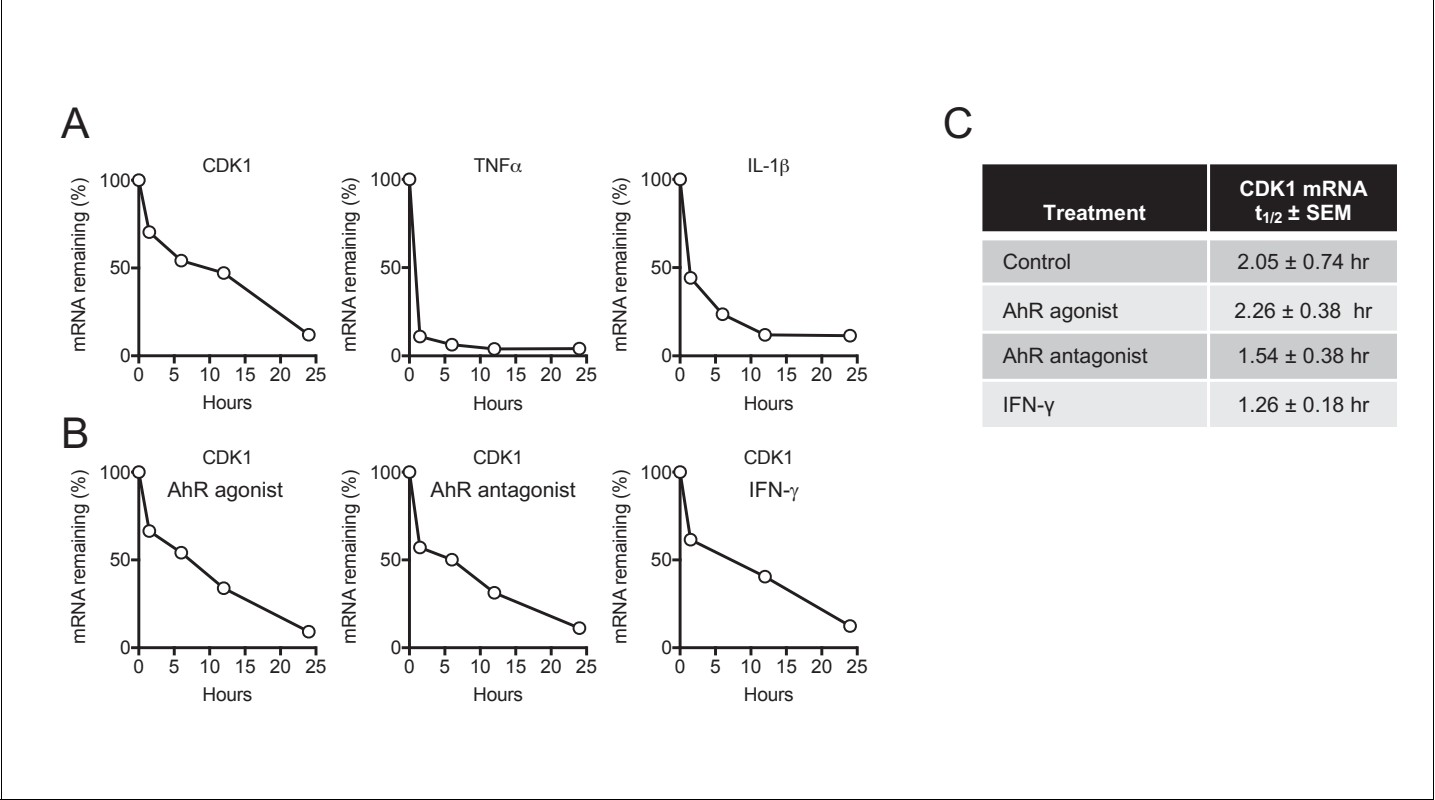

**Figure 7.** CDK1 mRNA stability is not affected by AhR and IFN-γ. (A) Decay of CDK1, TNFα and IL-1β following transcriptional blockage with actinomycin D, monitored over time by qPCR and normalized to GAPDH. Representative example from three different donors. (B) Decay rate of CDK1 mRNA following transcriptional blockage with actinomycin D in the presence of AhR agonist, antagonist or IFN-γ. CDK1 mRNA was monitored over time by qPCR and normalized to GAPDH. Representative example from three different donors. (C) Mean CDK1 mRNA half-life ($t_{1/2}$)±SEM for three different macrophage donors.

DOI: https://doi.org/10.7554/eLife.38867.014

## Discussion

Herein we describe an 'antiviral state' in human macrophages that results from AhR agonist or IFN-γ treatment via convergent mechanisms that involve transcriptional repression (*Figure 8*). While both AhR and IFN-γ are known to increase the expression of a number of target genes, we found that their propensity to repress expression of certain genes was key for their antiviral activity. Specifically, in both cases, transcriptional repression of CDK1, CDK2, and associated cyclins, reduces the degree to which SAMHD1 is phosphorylated and activates its dNTPase activity, thereby reducing the availability of dNTPs that are required for viral DNA synthesis (*Figure 8*).

AhR is recognized as influencing immune responses to extracellular and intracellular bacteria. Indeed, It has recently been found to be a pattern recognition receptor for pigmented bacterial virulence factors (phenazines and napthoquinones) (*Moura-Alves et al., 2014*). AhR deficiency thus increases mouse susceptibility to *Pseudomonas aeruginosa* and *Mycobacterium tuberculosis* infections. Mechanistically, AhR may influence immune responses to bacteria in multiple distinct ways. For example, AhR appears to facilitate the elicitation of anti-bacterial immune responses by regulating the production of IL-22 and other cytokines production by Th17 cells (reviewed by *Gutiérrez-Vázquez and Quintana, 2018*). AhR also appears to regulate the tissue distribution of lymphocytes (*Li et al., 2011*). An additional modulatory function of AhR provides protection against immunopathology by enhancing Treg cell differentiation and cytokine production as well as downregulating inflammation-associated gene expression in dendritic cells to promote 'disease tolerance' (*Gandhi et al., 2010*; *Apetoh et al., 2010*; *Bessede et al., 2014*).

Unlike bacteria, viruses are not thought to generate AhR ligands, therefore AhR has not frequently been studied in the context of viral infection. The few prior studies of AhR and viral infection

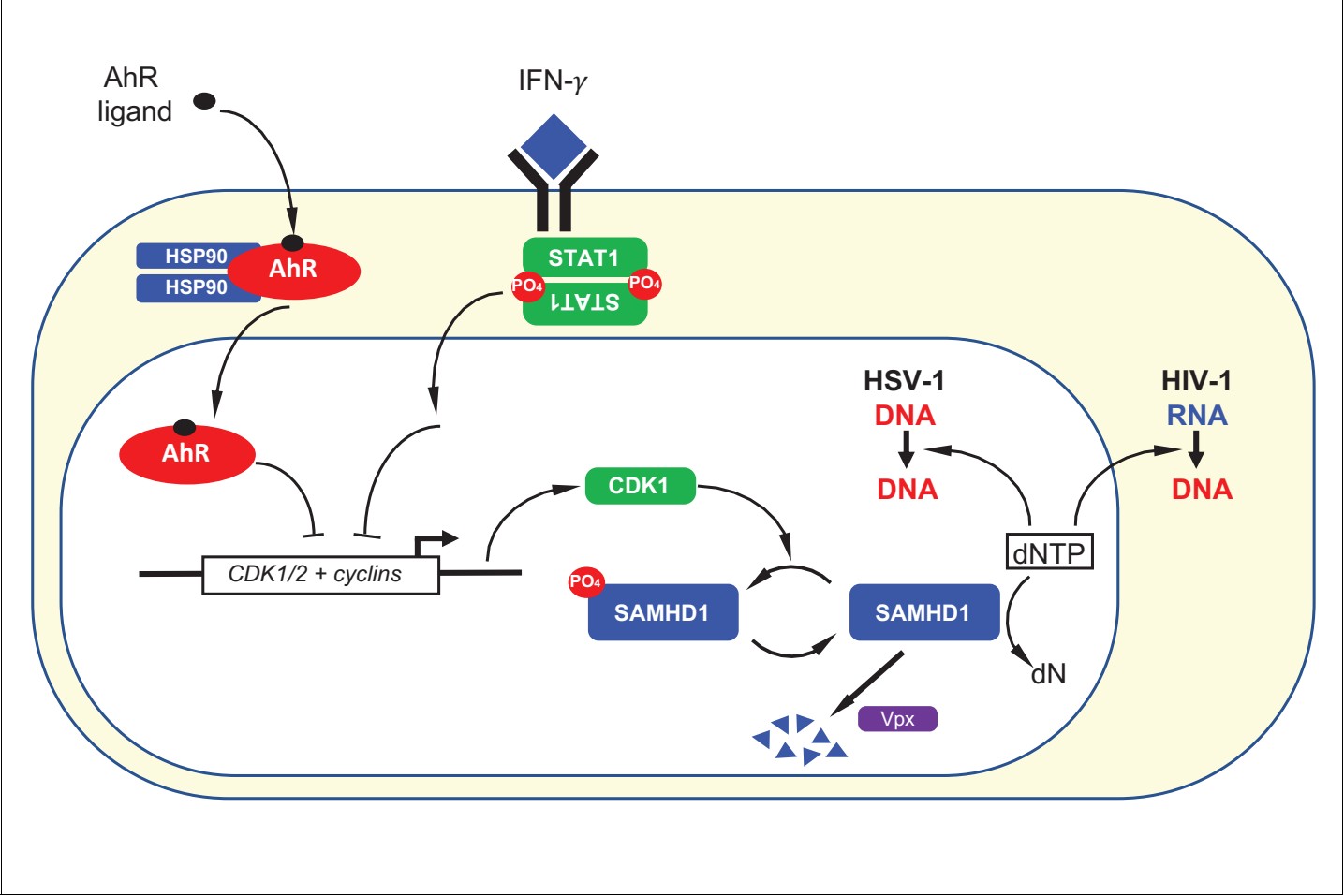

**Figure 8.** Summary depiction of the convergent mechanisms by which AhR agonists and IFN-γ inhibit on HIV-1 and HSV-1 replication. AhR agonist and IFN-γ mediated signaling both cause transcriptional repression of CDK1/2 and associated cyclins. This repression results in a reduction in SAMHD1 phosphorylation and, therefore, increased SAMHD1 catalytic activity. Catalytically active SAMHD1 reduces dNTP levels, thereby reducing HIV-1 and HSV-1 DNA synthesis. The SIV Vpx protein causes SAMHD1 degradation, ameliorating the antiviral effects of AhR ligands and IFN-γ.
DOI: https://doi.org/10.7554/eLife.38867.015

have employed 2,3,7,8-tetrachlorodibenzo-ρ-dioxin (TCDD), an environmental pollutant. While, historically, TCDD was frequently used as a prototypic AhR ligand, it can cause chronic and abnormal AhR activation due to its resistance to degradation by xenobiotic enzymes. In this context, AhR activation during viral infections was reported to exacerbate pathogenesis. For example, TCCD enhanced morbidity and mortality in mice and/or rats infected with influenza A viruses (*Lawrence and Vorderstrasse, 2013*), Coxsackievirus (*Funseth et al., 2002*) or following ocular HSV infection (*Veiga-Parga et al., 2011*). Exacerbation of viral infection may be related to the fact that AhR activation constrains the type-I interferon response (*Yamada et al., 2016*). Other studies have reported that TCDD triggers AhR-dependent HIV-1 gene expression in cell lines, but there has been conflicting data on which HIV-1 promoter elements are responsible (reviewed by *Rao and Kumar, 2015*).

While viruses are not thought to generate AhR ligands, AhR could nevertheless be activated in macrophages, other myeloid lineage cells or even other cell types, in virus infected individuals through a variety of mechanisms. The array of natural AhR ligands that are known to exist is increasing steadily in number and currently includes ligands provided by diet, commensal microbiota and tryptophan metabolism (*Stockinger et al., 2014*; *Zhang et al., 2017*). For example, during HIV-1 infection, CD4+ T cells are rapidly depleted, especially in gut associated lymphoid tissue (GALT). As a likely consequence, the gut microbiome of HIV-1 infected individuals are distinct in composition

from healthy individuals (*Bandera et al., 2018*; *Zilberman-Schapira et al., 2016*). Furthermore, dysbiosis and T cell depletion in HIV-1 infected patients may lead to the breakdown of the intestinal barrier, leading to the systemic distribution of bacterial products, as evidenced by increased circulating lipopolysaccharide (LPS) levels (*Brenchley et al., 2006*). It is plausible, even likely, that increased gut permeability would lead to the increased dissemination of bacteria-derived AhR ligands.

IDO1 may be another source of AhR ligands. IFN-γ treatment of macrophages dramatically upregulates expression of IDO1, and the major pathway of tryptophan metabolism is controlled by IDO1 as well as tryptophan 2,3-dioxygenase (TDO), both of which generate the metabolite kynurenine (*Stockinger et al., 2014*). Kynurenine is a low affinity AhR agonist and its role as AhR activator under physiological conditions unclear (*Opitz et al., 2011*). Indeed, in our experiments, we found no evidence for IFN-γ-induced, IDO1-mediated, AhR activation, suggesting the absence of crosstalk between IFN-γ and AhR antiviral pathways in human macrophages. However, a caveat to this notion is that our experiments were carried out in monolayered cell cultures. It is possible that in the context of a human tissue, with accompanying higher cell densities, that IDO1 metabolites could reach levels required to elicit an AhR response. Indeed, under some circumstances, kynurenine can be produced in large quantities and can drive AhR-mediated suppression of anti-tumor immune responses (*Opitz et al., 2011*). The physiological AhR ligand used herein (FICZ) is formed by photolysis of tryptophan and has an affinity for AhR similar to that of TCDD. FICZ is found particularly in the skin, but is also detected in human urine.

Our results demonstrate that AhR activation by this natural AhR ligand results in an inhibition of viral replication in macrophages. The effect of AhR and IFN-γ on HIV-1 replication was clearly cell-type specific, in that the large inhibitory activities seen in macrophages were greatly diminished or absent in CD4+ T cells. The propensity of AhR ligands and IFN-γ to deplete dNTPs should be determined by levels of AhR and IFN-γ receptor, as well as SAMHD1 levels and dNTP concentrations. Notably, AhR, IFN-γ receptor and SAMHD1 are expressed in many tissues and cell types, but SAMHD1 is especially abundant in myeloid and lymphoid cells and is induced by IFN-γ in some cell types (*Li et al., 2000*; *Liao et al., 2008*). Levels of dNTPs are determined both by dNTP degrading SAMHD1 and synthesizing (e.g. ribonucleotide reductase) enzymes, and are unlikely to be sufficiently low in actively dividing cells for the antiviral activities documented herein to be exerted (*Amie et al., 2013*). Rather, non-dividing cells with intrinsically low dNTP levels are likely to provide situations in which dNTP-lowering antiviral strategies are effective. Prior work has demonstrated that dNTPs are 20- to 300-fold lower in macrophages compared to activated PBMC (*Diamond et al., 2004*; *Amie et al., 2013*). However, the levels of dNTPs in terminally differentiated or non-dividing cells and in other in vivo tissues is surprisingly poorly characterized (*Amie et al., 2013*). Thus, further experimentation will be required to determine whether the antiviral activities of AhR and IFN-γ documented herein are widespread among tissues or confined to cells of the myeloid lineage.

Unfortunately, delivery of reporter constructs into macrophages via various methods so altered their responsiveness to AhR ligands and IFN-γ that we were unable to probe effects of AhR activation and IFN-γ treatment on CDK1 promoter activity. Nevertheless, our finding that CDK1 mRNA turnover was unaltered by AhR and IFN-γ, while steady state mRNA levels were reduced by >10 fold indicates that the rate of CDK1 mRNA synthesis must be reduced by AhR ligands and IFN-γ. Further work will be required to determine precisely how transcriptional repression is achieved. Previous work has suggested that p27$^{Kip1}$ can be induced by TCDD-mediated AhR activation, resulting in cell cycle arrest (*Kolluri et al., 1999*), while other studies demonstrated that TCDD treatment of human or mouse cancer cell lines resulted in the recruitment of AhR to E2F-dependent promoters with subsequent repression through a mechanism involving displacement of p300 (*Marlowe et al., 2004*). It is possible that these affects are related to our findings of CDK/cyclin downregulation, particularly since the CDK1 promoter contains functional E2F binding sites (*Dalton, 1992*; *Sugarman et al., 1995*; *Tommasi and Pfeifer, 1995*). IFN-γ has been shown to induce a cell cycle arrest in various tumor cells by multiple cell type-dependent pathways, including via upregulation of cyclin-dependent kinase inhibitors p21$^{WAF1/Cip1}$ and p27$^{Kip1}$, the signal transducer and activator of transcription-1 (STAT1) signaling pathway or through the activation of E2F transcription factor proteins via p21$^{WAF1/Cip1}$ (*Schroder et al., 2004*). Further work will determine how these phenomena are related to the downregulation of CDK/cyclin expression documented herein.

While IFN-γ is largely thought of as an immunoregulatory cytokine, some reports have documented antiviral activity against various viruses, including HIV-1. However, the mechanisms

underlying antiviral activity have largely proven elusive (*Rihn et al., 2017*). IFN-γ has been shown to play an important role in HSV-1 pathogenesis (*Bigley, 2014*), but the mechanisms by which it inhibits virus replication is seen as largely via largely through modulation of innate and adaptive cellular responses rather than directly affecting viral replication. For HIV-1, IFN-γ has been shown to be directly antiviral, and cause both early and late blocks to replication. While the early block was poorly characterized, late blocks have been shown to result in part from IDO1 dependent trypto-phan starvation (*Kane et al., 2016*). Another IFN-γ imposed late block was shown to be cell-type and viral strain-dependent, with viral determinants mapping to the *env* gene (*Rihn et al., 2017*). It is likely that previously described early blocks, that were observed in monocytoid cell lines, are mecha-nistically the same at that described herein for primary macrophages.

A number of different stimuli or signals, including the DNA damage response (*Mlcochova et al., 2018*), AhR ligands and IFN-γ signaling appear to converge on dephosphorylation of SAMHD1. Indeed, immediately before submission of this manuscript another group reported that Type-I, II and III interferon can induce SAMHD1 dephosphorylation (*Szaniawski et al., 2018*). In principle, each of these signaling pathways could be activated as a consequence of infection by retroviruses or DNA viruses. Thus, dephosphorylation of SAMHD1 and consequent dNTP depletion may be a common strategy to limit retrovirus or DNA virus replication. As such targeted activation of AhR, or CDK downregulation via other means may represent a viable antiviral therapeutic strategy.

## Materials and methods

### Cells

Fresh human peripheral blood mononuclear cells (PBMCs) were isolated by Ficoll-Paque gradient centrifugation, and plated in serum-free medium for 3 hr at 37°C. The non-adherent cells were dis-carded and adherent monocytes were cultured in RPMI (Gibco) with 10% FCS (Sigma) and recombi-nant human granulocyte macrophage colony-stimulating factor (GM-CSF) (100 ng/ml, Gibco). Cell differentiation to macrophages was completed after for 4 to 6 days, and cells were treated as indi-cated at day seven prior to infection.

Primary CD4+ T cells were isolated from human blood by Ficoll-Paque gradient centrifugation and negative selection (RosetteSep Human CD4+ T Cells Enrichment Cocktail, StemCell Technolo-gies). Primary CD4+ T cells and PBMCs were activated with Phytohemagglutinin-L (2 µg/ml, Sigma) for 48 hr, cultured in the presence of interleukin-2 (50 U/ml, PeproTech), and treated as indicated for 16 hr prior to infection.

### Viruses

The infectious proviral DNA clone HIV-1$_{NLYU2}$, as well as derivatives encoding GFP or nluc encoding (in place of *nef*) were used as previously described (*Zhang et al., 2002*). The HIV-1$_{89.6}$ infectious molecular clone of the primary HIV-1 isolate 89.6 was obtained from the NIH AIDS Reagents Pro-gram. The SHIV AD8 provirus was kindly provided by Theodora Hatziioannou. SHIV AD8 ΔVpx was generated by mutating residues T2C, G175T, C178T and deleting nucleotide G183 resulting in a mutated start codon, the introduction of 2 stop codons and a frame shift respectively. Vpx-contain-ing and control SIV$_{mac251}$ constructs, SIV3+ and SIV3+ ΔVpx respectively, were kindly provided by Caroline Goujon. An HSV-1 (strain KOS) virus stock was kindly provided by Margaret MacDonald.

For the generation of HIV-1 virus stocks, 293T cells in a 10 cm plate were transfected with 10 µg of proviral plasmid DNA using polyethyleneimine (PolySciences) and placed in fresh medium after 24 hr. At 40 hr post-transfection, virus-containing cell supernatant was harvested, treated with 100 U of DNase I (Roche) for 1 hr at 37°C in the presence of 10 mM MgCl$_2$, concentrated using Lenti-X Con-centrator (Takara), resuspended in RPMI medium without serum and stored at −80°C. To determine infectious HIV-1 and SHIV titers, serial dilutions of the virus stocks were used to infect TZM reporter cells as previously described by (*Bitzegeio et al., 2013*).

SIV3+ and SIV3+ ΔVpx virus like particles (VLPs) were produced as described by *Goujon et al.* (*2013*). Virus containing supernatants were treated with 100 U of DNase I (Roche) for 1 hr at 37°C in the presence of 10 mM MgCl$_2$, purified by ultracentrifugation through a sucrose cushion (20% w/v; 75 min; 4°C; 28,000 rpm), resuspended in RPMI medium without serum and stored at −80°C. The

amount of VLPs was normalized using a one-step SYBR Green I based PCR-based reverse transcriptase (RT) assay, as described previously (*Del Prete et al., 2017*).

## HIV-1 and SHIV spreading replication assays

For replication assays in macrophages, $1.5 \times 10^5$ macrophages were plated in each well of a 24-well plate and treated with 0.3% DMSO (Sigma), 0.5 µM 6-Formylindolo(3,2-b)carbazole (FICZ) (Enzo) or 3 µM CH-223191 (Sigma) or the indicated concentrations (in U/ml) of IFN-γ (Gibco) for 16 hr. Thereafter cells were infected with $1.5 \times 10^5$ IU (measured on TZM-bl cells) of HIV-1$_{NLYU2}$, HIV-1$_{89.6}$, or SHIV AD8. At 6 hr post infection, cells were washed and supernatants collected over 8 days. The amount of virus released into the supernatant was quantified using a one-step SYBR Green I based PCR RT assay, as described (*Del Prete et al., 2017*).

For replication assays in lymphocytes, $1 \times 10^4$ PHA-activated PBMCs or CD4 +T cells were plated in one well of a 96-well plate, treated as described above, infected with $1 \times 10^3$ IU of HIV-1$_{NLYU2}$, HIV-1$_{89.6}$, or SHIV AD8. Supernatants were collected and analyzed as described above. U937 CD4/R5 cells expressing candidate genes were infected in the same way except that cells were treated with 50 ng/ml Phorbol 12-myristate 13-acetate (PMA) (Sigma) 2 days prior to infections.

## Measurement of HSV-1 replication

For HSV-1 replication experiments, $8 \times 10^5$ macrophages were treated with 0.3% DMSO, 0.5 µM FICZ, 3 µM CH-223191 or 1000 U/ml IFN-γ for 16 hr, followed by infection with HSV-1 at an MOI of 2. At 1 hr post infection, cells were washed, supernatants and cells were collected at 12 and 24 hr post infection. HSV-1 titers were determined as described (*Bick et al., 2003*).

## Incoming HIV-1 infection assay

Macrophages were plated in one well of a 96-well plate, treated with SIV3 +or SIV3+ ΔVpx virus like particles for 10 hr, followed by treatment with 0.3% DMSO, 0.5 µM FICZ, 3 µM CH-223191 or 1000 U/ml of IFN-γ; or indicated amount of L-Tryptophan and 1-Methyl-L-tryptophan (Sigma) for 16 hr before infection with HIV-1$_{NLYU2}$ nluc. Cells were lysed 48 hr post infection and luciferase was measured using the Nano-Glo Luciferase Assay System (Promega) and Modulus II Microplate Multimode Reader (Turner BioSystems).

## Measurement of HIV-1 DNA species in infected cells

For analysis of HIV-1 late reverse transcription products in infected cells, $8 \times 10^5$ macrophages were seeded in 6-well culture plates and infected with $8 \times 10^6$ IU (as determined on TZM-bl cells) of HIV-1$_{NLYU2}$. Cells were either treated with 0.3% DMSO or 0.5 µM FICZ for 16 hr prior to infection, or were treated with Nevirapine (Sigma) starting at the time of infection. Total DNA was extracted 24 hr post infection, using the NucleoSpin Tissue Kit (Macherey-Nagel). The resulting DNA samples were used as template for quantitative PCR using TaqMan Gene Expression Master Mix (Applied Biosystems) and StepOne Plus Real-Time PCR system (Applied Biosystems). The primer pairs and PCR conditions used in this study were as described previously (*Butler et al., 2001*).

## Microarray analysis

Total RNA was extracted, using the RNeasy Plus Mini Kit (QIAGEN), from human macrophages that were treated with 0.3% DMSO (Sigma), 0.5 µM 6-Formylindolo(3,2-b)carbazole (alternative name FICZ) (Enzo) or 3 µM CH-223191 (Sigma) for 24 hr before harvest. Alternatively, macrophages were either untreated or treated with 1000 U/ml IFN-γ (Gibco) for 24 hr before harvest. Complementary RNA was prepared and probed using Human HT-12 v4.0 Gene Expression BeadChips (Illumina), according to manufacturer's instructions.

## Stable expression of individual genes

The U937 cell line was purchased from the ATCC. A U937-derived cell line expressing human CD4 and CCR5 as a single transcript linked by a self-cleaving picornovirus-derived 2A peptide-encoding sequence was generated by transduction with a pLHCX-derived vector followed by selection with 50 µg/ml Hygromycin B. A single cell clone was derived from the population of transduced cells by limiting dilution. Cell identity was inferred by visual inspection and response to PMA (differentiation to

an adherent phenotype). Cells were not tested for mycoplasma contamination, but were periodically tested for retrovirus contamination using a PCR-based reverse transcriptase assay. For constitutive expression of genes induced in the AhR and IFN-γ microarrays, a lentiviral vector, CSIB, was derived from CSGW by replacing sequences encoding GFP with a multi-cloning site followed by an IRES sequence and a blasticidin resistance cassette. A selection of genes that were induced by 0.5 µM FICZ or 1000 U/ml IFN-γ were cloned into CSIB and were introduced into the U937 cells with CSIB based vectors followed by selection with 5 µg/ml blasticidin. A control cell line containing empty vector CSIB was similarly generated. The genes analyzed by this method were GPR183, CXCR5, ADK, COL23A1, CYGB, RASAL1, NCF1, SEMA6B, TNFAIP8L3, SLC7A11, XYLT, FUCA1, LRG1, ITGB7, ANKRD22, CCL7, CD40, FCGR1A, GIMAP8, FCGR1B, FGL2, RABGAP1L, PST-PIP2, PSME2, GIMAP6, HAPLN3, IL27, LYSMD2, MAP3K7CL, NFIX, NUPR1, GPBAR1, TMEM110, UBD, C5, SCIMP, SDS, SOD2, STAMBPL1, CIITA, JUP, PARP14 and METTL7B.

## Measurement of mRNA levels using PCR

Macrophages ($8 \times 10^5$) were treated with 0.3% DMSO, 0.5 µM FICZ, 3 µM CH-223191 or 1000 U/ml IFN-γ for 24 hr, and RNA was isolated using the NucleoSpin RNA Kit (Macherey-Nagel). cDNA was synthesized with SuperScript III First-Strand Synthesis System (Invitrogen), and analyzed by quantitative PCR using TaqMan Gene Expression Master Mix (Applied Biosystems). Duplicate reactions were run according to manufacturer's instructions using a StepOne Plus Real-Time PCR system (Applied Biosystems). For relative quantification, samples were normalized to GAPDH. The following TaqMan Gene Expression Assays (Applied Biosystems) were used: GAPDH (Hs02786624_g1), CYP1B1 (Hs00164383_m1), CDK1 (Hs00938777_m1), CDK2 (Hs01548894_m1), CCNA2 (Hs00996788_m1), CCNE2 (Hs00180319_m1).

## Western blotting

Cells were lysed in LDS sample buffer (Invitrogen), separated by electrophoresis on NuPage 4–12% Bis-Tris gels (Invitrogen) and blotted onto nitrocellulose membranes (GE Healthcare). Membranes were incubated with rabbit anti-HSP90 (Santa Cruz Biotechnology), mouse anti-SAMHD1 (OriGene), rabbit anti-pSAMHD1 (ProSci), mouse anti-cdc2 (Cell Signaling), rabbit anti-CDK1 (proteintech), rabbit anti-CDK2 (proteintech), rabbit anti-CCNE2 (proteintech), rabbit anti-cyclin A2 (proteintech), rabbit anti-STAT1 (proteintech), rabbit anti-pSTAT1 (Cell Signaling), mouse anti-HSV-1 gD (Santa Cruz Biotechnology) or mouse anti-IDO1 (proteintech). Thereafter, membranes were incubated with goat anti-rabbit IRDye 800CW and goat anti-mouse IRDye 680RD (respectively LI-COR Biosciences), and scanned using a LI-COR Odyssey infrared imaging system. Alternatively, membranes were incubated with appropriate horseradish peroxidase (HRP) conjugated secondary antibodies (Jackson ImmunoResearch), and visualized using SuperSignal West Femto Chemiluminescent solution (Thermo Fisher) and a C-DiGit Western Blot Scanner.

## Whole-cell dNTP quantification

Macrophages were plated at $5 \times 10^6$ cells per 10 cm dish, treated with 0.3% DMSO, 0.5 µM FICZ, 3 µM CH-223191 or 1000 U/ml IFN-γ for 24 hr, washed in 1x PBS and lysed in 65% methanol. Whole-cell dNTP contents were measured using a single nucleotide incorporation assay, as described (*Diamond et al., 2004*).

## Measurement of mRNA stability

Macrophages ($6 \times 10^5$) were treated with 5 µg/ml actinomycin D (Sigma) in combination with 0.3% DMSO, 0.5 µM FICZ, 3 µM CH-223191 or 1000 U/ml IFN-γ for indicated amounts of time and mRNA levels were measured as described above. As controls, the following additional TaqMan Gene Expression Assays (Applied Biosystems) were used: TNF (Hs00174128_m1) and IL1B (Hs01555410_m1).

## Phenotypic analysis of macrophages by flow cytometry

For analysis of macrophage marker expression, macrophages were harvested in Accutase (Sigma), pelleted and washed in FACS buffer (PBS, 0.2% BSA). Macrophages were then incubated with Human Fc Block (BD Biosciences), and stained with the following antibodies (all from BD

Biosciences): anti-CD14 (555397), CD206 (550889), CD80 (561135) and CD163 (556018) or each relevant isotype control. All data were acquired on an Attune cytometer (Thermo Scientific) and analyzed with FlowJo software.

### Cell-cycle analysis

Macrophages were harvested in Accutase (Sigma), pelleted and washed in PBS and stained with FxCycle PI/RNase Staining Solution (Thermo Scientific) according to manufacturer's instructions. DNA content was then measured by FACS analysis using an Attune cytometer (Thermo Scientific) and analyzed with FlowJo software.

## Acknowledgments

We thank Theodora Hatziioannou for the SHIV AD8 proviral plasmid, Caroline Goujon for SIV3+ constructs, and NIH AIDS reagents program for HIV-1 infectious molecular clones, João Lucas Duarte, Gitta Stockinger and members of the Bieniasz and Hatziioannou labs for helpful discussions and Margaret MacDonald for the HSV-1 KOS virus stock. This work was supported by grants from the NIH R37AI64003 (to PDB), R01 GM104198 (to BK) and R01 AI136581 (to BK). The content of this manuscript is solely the responsibility of the authors and does not necessarily represent the official views of the National Institutes of Health.

## Additional information

### Funding

| Funder | Grant reference number | Author |
|---|---|---|
| Howard Hughes Medical Institute | | Paul D Bieniasz |
| National Institute of Allergy and Infectious Diseases | R37AI64003 | Paul D Bieniasz |
| National Institute of General Medical Sciences | R01 GM104198 | Baek Kim |
| National Institute of Allergy and Infectious Diseases | R01 AI136581 | Baek Kim |

The funders had no role in study design, data collection and interpretation, or the decision to submit the work for publication.

### Author contributions

Tonya Kueck, Conceptualization, Formal analysis, Supervision, Validation, Investigation, Methodology, Writing—original draft, Writing—review and editing; Elena Cassella, Jessica Holler, Investigation; Baek Kim, Supervision, Methodology, Project administration, Writing—review and editing; Paul D Bieniasz, Conceptualization, Formal analysis, Supervision, Funding acquisition, Writing—original draft, Project administration, Writing—review and editing

### Author ORCIDs

Paul D Bieniasz (iD) http://orcid.org/0000-0002-2368-3719

### Decision letter and Author response

Decision letter https://doi.org/10.7554/eLife.38867.023
Author response https://doi.org/10.7554/eLife.38867.024

# Additional files

## Supplementary files

• Supplementary file 1. Top 100 upregulated genes in AhR activated macrophages. The fold change in mRNA levels, relative to carrier treated cells, for the top 100 AhR-activation-induced genes is given.

DOI: https://doi.org/10.7554/eLife.38867.016

• Supplementary file 2. Top 100 downregulated genes in AhR activated macrophages. The fold change in mRNA levels, relative to carrier treated cells, for the top 100 AhR-activation-repressed genes is given.

DOI: https://doi.org/10.7554/eLife.38867.017

• Supplementary file 3. Pathway analysis of top 100 downregulated genes in (i) AhR activated macrophages and (ii) IFN-γ stimulated macrophages

DOI: https://doi.org/10.7554/eLife.38867.018

• Supplementary file 4. Top 100 upregulated genes in IFN-γ treated macrophages. The fold change in mRNA levels, relative to carrier treated cells, for the top 100 IFN-γ-induced genes is given.

DOI: https://doi.org/10.7554/eLife.38867.019

• Supplementary file 5. Top 100 downregulated genes in IFN-γ treated macrophages. The fold change in mRNA levels, relative to carrier treated cells, for the top 100 IFN-γ-repressed genes is given.

DOI: https://doi.org/10.7554/eLife.38867.020

• Transparent reporting form

DOI: https://doi.org/10.7554/eLife.38867.021

## Data availability

All data generated or analysed during this study are included in the manuscript and supporting files.

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
