## [Decision Letter]

Thank you for submitting your article "The aryl hydrocarbon receptor and interferon gamma generate antiviral states via transcriptional repression" for consideration by *eLife*. Your article has been reviewed by Arup Chakraborty as the Senior Editor, a Reviewing Editor, and two reviewers. The following individual involved in review of your submission has agreed to reveal his identity: Bryan Cullen (Reviewer #2).

The reviewers have discussed the reviews with one another and the Reviewing Editor has drafted this decision to help you prepare a revised submission.

Summary:

This interesting and well-written manuscript first shows that agonists specific for the aryl hydrocarbon receptor (AhR) are potent inhibitors of HIV replication in macrophages but not T cells. They then show that this effect results from reduced expression of CDKs which in turn results in reduced phosphorylation of the known restriction factor SAMHD1, which increases SAMHD1 activity and depletes dNTP levels. Consistent with this model, AhR agonists also inhibit HSV-1 replication and SIV Vpx, a known inhibitor of SAMHD1 activity, rescues the inhibitory phenotype. The authors then go on to show that interferon gamma, which is known to inhibit HIV replication in macrophages but not T cells, also reduces the expression of the same CDKs and exerts a very similar phenotype via activation of SAMHD1. These data explain why interferon gamma acts as a tissue specific inhibitor of HIV replication and illuminate the mechanistic basis for this effect. This is an excellent manuscript, which could be improved by addressing some specific points as below.

Essential revisions:

1) Where representative experiments alone are shown, statistical analyses would be useful to demonstrate reproducibility, and to address the issue of the variability of results from donor to donor, as already shown in some cases. Phenotypic analyses of cell differentiation of monocyte to macrophages may also help explain variability between experiments.

2) The authors use total PBMCs to demonstrate the absence of regulation of CDK1/2 protein levels upon AhR activation (Figure 2—figure supplement 3A) and SAMHD1 phosphorylation state (Figure 2—figure supplement 3A). Nonetheless, since PBMCs are a mix of various cell types, this diversity might mask putative regulation in the relevant cell type susceptible to HIV infection. Similar analysis using the CD4^+^ T cells, where HIV replication is not modulated by AhR agonists/antagonists, would be more appropriate.

3) A schematic representation of the key findings integrated into a working model of the regulatory signaling and cellular factors involved would further highlight the impact of this study.

---

## [Author Response]

Essential revisions:1) Where representative experiments alone are shown, statistical analyses would be useful to demonstrate reproducibility, and to address the issue of the variability of results from donor to donor, as already shown in some cases. Phenotypic analyses of cell differentiation of monocyte to macrophages may also help explain variability between experiments.

We have added statistical and collated analyses for all experiments that were previously presented only in the form of representative data. Specifically, we have added additional panels presenting collated data with statistics in Figure 1—figure supplement 1 (panel I), Figure 3—figure supplement 1 (panel D) and Figure 7 (panel C). Additionally, Figure 4 panel C and Figure 6 panel D that previously displayed representative experiments, now show collated data with statistics in the revised manuscript.

Unfortunately, our macrophage preparation procedure does not include an intermediate monocyte isolation step that would conveniently enable us to monitor cells pre- and post- differentiation. Nevertheless, in the revised manuscript we have included a panel showing phenotypic analyzed macrophages with commonly used macrophage markers (now shown in Figure 1—figure supplement 2 (panel A)).

2) The authors use total PBMCs to demonstrate the absence of regulation of CDK1/2 protein levels upon AhR activation (Figure 2—figure supplement 3A) and SAMHD1 phosphorylation state (Figure 2—figure supplement 3A). Nonetheless, since PBMCs are a mix of various cell types, this diversity might mask putative regulation in the relevant cell type susceptible to HIV infection. Similar analysis using the CD4^+^ T cells, where HIV replication is not modulated by AhR agonists/antagonists, would be more appropriate.

We concede this point and have revised the manuscript to include Western blot analysis of CDK1/2 protein levels and phospho-SAMHD1 levels in purified CD4 + T cells, in addition to the previous data obtained using unfractionated PBMC. The new data is presented as (Figure 2—figure supplement 1B). Notably, the purified CD4^+^ T cells showed the same lack of effect of AhR on CDK levels and SAMHD1 phosphorylation as did the unfractionated PBMC.

3) A schematic representation of the key findings integrated into a working model of the regulatory signaling and cellular factors involved would further highlight the impact of this study.

We have included a schematic representation of the key findings as a new Figure 8.